# The genome polishing tool POLCA makes fast and accurate corrections in genome assemblies

**Aleksey V. Zimin**[1,2]*, **Steven L. Salzberg**[1,2,3,4]

**1** Department of Biomedical Engineering, Johns Hopkins University, Baltimore, Maryland, United States of America, **2** Center for Computational Biology, Johns Hopkins University, Baltimore, Maryland, United States of America, **3** Department of Computer Science, Whiting School of Engineering, Johns Hopkins University, Baltimore, Maryland, United States of America, **4** Department of Biostatistics, Bloomberg School of Public Health, Johns Hopkins University, Baltimore, Maryland, United States of America

* alekseyz@jhu.edu

## Abstract

The introduction of third-generation DNA sequencing technologies in recent years has allowed scientists to generate dramatically longer sequence reads, which when used in whole-genome sequencing projects have yielded better repeat resolution and far more contiguous genome assemblies. While the promise of better contiguity has held true, the relatively high error rate of long reads, averaging 8–15%, has made it challenging to generate a highly accurate final sequence. Current long-read sequencing technologies display a tendency toward systematic errors, in particular in homopolymer regions, which present additional challenges. A cost-effective strategy to generate highly contiguous assemblies with a very low overall error rate is to combine long reads with low-cost short-read data, which currently have an error rate below 0.5%. This hybrid strategy can be pursued either by incorporating the short-read data into the early phase of assembly, during the read correction step, or by using short reads to "polish" the consensus built from long reads. In this report, we present the assembly polishing tool POLCA (POLishing by Calling Alternatives) and compare its performance with two other popular polishing programs, Pilon and Racon. We show that on simulated data POLCA is more accurate than Pilon, and comparable in accuracy to Racon. On real data, all three programs show similar performance, but POLCA is consistently much faster than either of the other polishing programs.

This is a *PLOS Computational Biology* Software paper.

## Introduction

Third-generation sequencing platforms such as Single Molecule Real Time (SMRT) sequencing by Pacific Biosciences (PacBio) and nanopore sequencing by Oxford Nanopore Technologies (ONT) yield reads that can range in size from a few kilobases to more than a megabase. However, both technologies have a relatively high error rate of 8–15%. The types of errors

**Data Availability Statement:** All relevant data are within the manuscript and its Supporting Information files.

**Funding:** This work was supported in part by the USDA National Institute of Food and Agriculture https://nifa.usda.gov/ under grant 2018-67015-

28199), by National Science Foundation https://
www.nsf.gov/ grant IOS-1744309, and by National
Institutes of Health https://www.nih.gov/ grants
R01-HG006677 and R35-GM130151. The funders
had no role in study design, data collection and
analysis, decision to publish, or preparation of the
manuscript.

**Competing interests:** The authors have declared
that no competing interests exist.

differ between technologies, but with sufficiently deep coverage, most errors can be corrected by using reads to cross-check each other. Another strategy for error correction is to pair the long-read rata with short (100–250bp) Illumina reads, which have error rates below 0.5%. The hybrid strategy requires significantly lower coverage by the more-expensive long reads, which can be replaced by much-cheaper Illumina reads. Using a second technology has the additional advantage that systematic errors in the long reads might not be corrected even with deep coverage, and the Illumina reads can be used to correct these errors. Whole-genome assemblies assembled using a hybrid sequence strategy can thereby obtain an overall error rate of less than 1 error per 100 thousand bases [1].

There are two ways one can use Illumina data in a hybrid genome project. One can either use it early in the process to correct long reads, as is done in the PBcR [2] and MaSuRCA [3] assemblers, or one can use it after the long read assembly has been completed to improve the quality of the consensus by aligning the Illumina reads to the assembly. This latter approach is commonly referred to as "polishing" the consensus. There are several software tools available for polishing assemblies with Illumina data, with the most widely used ones being Pilon [4] and Racon [5]. In this paper we present a novel polishing tool called POLCA (POLishing by Calling Alternatives), which we are distributing with the MaSuRCA assembler package starting with version 3.3.5. The current version of POLCA described in this paper is available in MaSuRCA version 3.4.1. POLCA has three main advantages over the widely used tools Pilon and Racon: (1) it is very fast, (2) it uses very little memory, and (3) it makes more accurate corrections. As our experiments demonstrate, the polished sequence quality is better than the quality achieved by either Pilon or Racon. We also compare POLCA to two newer tools, ntEdit [6] and NextPolish [7]. Compared to the new tools, POLCA has comparable performance to NextPolish and it outperforms ntEdit by wide margin. Its speed, accuracy, and ease of use make POLCA a good tool for assembly polishing.

In the following we present our analysis of POLCA's performance on three data sets. First, we use a simulated data set where we introduce known random errors into a genome and polish it with reads simulated from the same genome. This lets us compare the polished assembly to the "true" genome sequence. We then test our polishing methods on a set of bacterial genome assemblies produced from Oxford Nanopore data, and on a human genome assembled from PacBio data.

## Design and implementation

There are at least two approaches to polishing the consensus sequence of an existing assembly. One is to recover the multi-alignment of the reads by aligning them to the genome assembly, and then re-doing the consensus calculation using the original or additional read data. A second approach is to align the reads to the consensus, identify any locations where the reads indicate a possible error, and then to fix those errors using the read sequences. The first approach, which is followed by Racon and Pilon, is more computationally expensive, but it may work better when assemblies contain a large number of errors. POLCA employs the latter approach.

POLCA is implemented as a bash script program that takes as input a file of Illumina reads and the target assembly to be polished. The outputs are the polished assembly and a VCF (variant call format) file containing the variants used for polishing. The basic outline of the script is to align the Illumina reads to the genome and then call short variants from the alignments. A variant call is treated as a putative error in the consensus if the count of the alternative allele observations is greater than 1 and at least twice the count of the reference allele. Each error is fixed by replacing the error variant with the highest scoring alternative allele suggested by the Illumina reads. The variants can be substitutions or insertions/deletions of one or more bases.

POLCA uses bwa mem [8] to align reads to the assembly, but another short-read aligner can easily be substituted. For variant calling, it uses FreeBayes [9] due to its stability and portability; however, by default FreeBayes can only use a single thread (processor). In POLCA we use shell level multiprocessing FreeBayes to run multiple instances of FreeBayes in parallel, thus significantly speeding up the variant calling. We also tuned its alignment and variant calling parameters to improve sensitivity, specificity, and speed for detecting consensus errors. The FreeBayes binary is included with the POLCA distribution as part of the MaSuRCA package. (Note that POLCA installs with MaSuRCA but can be run independently to polish assemblies produced with third-party assemblers.)

POLCA first builds an index of the target assembly, and then aligns the Illumina reads to the target with bwa. It then uses samtools to sort the alignment (bam) file. For variant calling we run FreeBayes in 5Mb batches, merging the variant call vcf files after all batches finish. We then process the assembly using the computed variant calls in parallel, where the number of batches is equal to the user-specified number of CPUs. We extract all target sequence names, sort them in lexicographic order and split the sorted list into batches. This helps balance the amount of target sequence in each batch, thus balancing the load on the CPUs. Parallel execution is achieved using the "xargs -P" command, which ensures compatibility between different Unix-based systems.

## Results

To evaluate POLCA, we compared its performance to two widely used genome polishing tools, Pilon and Racon. We compared using three data sets: first, a simulated data set with Illumina-like reads based on the *Arabidopsis thaliana* genome, with simulated errors introduced into the genome sequence. The second experiment used a published human NA12878 assembly, sequenced and assembled from Pacific Biosciences SMRT data and available as GenBank accession GCA_001013985.1 [10]. The third experiment used several *Klebsiella pneumoniae* bacterial genomes sequenced with both Oxford Nanopore and Illumina data in a study in [1].

### Simulated data experiments

The faux data set was based on the finished sequence of *A. thaliana* TAIR1.0 (GenBank accession GCA_000001735.1). We removed all N's and non-ACGT characters from the genome sequence and called this sequence the "clean" genome. We then set up three experiments where we introduced random errors into the clean genome with probability $e$ at each base. The errors themselves consisted of 90% substitutions, 5% insertions, and 5% deletions. The size of each insertion or deletion error was chosen uniformly at random from the range [1,20]. This ensured that approximately the same number of bases would appear in SNPs and indels (insertions or deletions). All substitutions were random bases differing from the true base; all insertions were random sequences of bases. The code for introducing errors into assembled genomes is included with the MaSuRCA package and its usage is described in the README. md file on GitHub. We created five simulated genomes with $e$ taking values 0.0002, 0.0005, 0.001, 0.0025 and 0.005, which translated into genomic consensus error rates of approximately 0.037%, 0.094%, 0.18%, 0.46%, and 0.92%. We then simulated 30x coverage of the clean genome in simulated 250bp ("Illumina") paired reads with a 1% error rate, using wgsim (https://github.com/lh3/wgsim) with parameters "-r 0 -e 0.01 -N 7200000–1 250–2 250". Note that the simulated Illumina reads had an error rate approximately twice as high as that observed in real Illumina reads.

**Table 1** compares the performance of POLCA to Pilon on a subset of three experiments; **Fig 1** shows the comparisons for all five simulated error rates. Both POLCA and Pilon report

**Table 1. Results for error correction by POLCA and Pilon on an *A. thaliana* genome (total size 119Mb) with three different numbers of simulated errors.** Error rates ranged from 0.1% to 0.46%. Boldface indicates the better values for each experiment in each row.

| | POLCA | | | Pilon | | |
|---|---|---|---|---|---|---|
| Experiment (error rate) | Exp 1 (0.1%) | Exp 2 (0.2%) | Exp 3 (0.46%) | Exp 1 (0.1%) | Exp 2 (0.2%) | Exp 3 (0.46%) |
| Simulated substitution errors | 53,726 | 107,244 | 267,896 | 53,726 | 107,244 | 267,896 |
| Substitutions fixed (TP) | 48,442 | 97,093 | 241,883 | **49,545** | **98,825** | **246,405** |
| Substitutions missed (FN) | 5,284 | 10,151 | 26,013 | **4,181** | **8,419** | **21,491** |
| Substitution errors introduced (FP) | **4** | **27** | **68** | 2,019 | 3,887 | 9,471 |
| Simulated indel errors | 57,758 | 112,894 | 281,332 | 57,758 | 112,894 | 281,332 |
| Indels fixed (TP) | 54,802 | **107,588** | **268,702** | 55,463 | 107,576 | 261,279 |
| Indels missed (FN) | 2,956 | **5,306** | **12,630** | 2,295 | 5,318 | 20,053 |
| Indel errors introduced (FP) | **237** | **708** | **1,560** | 2,796 | 5,543 | 19,177 |
| Total errors remaining after polishing | **8,481** | **16,192** | **40,271** | 11,291 | 23,167 | 70,192 |

all the corrections that they make, allowing us to evaluate the corrections precisely, computing the number of true positives (corrected errors, TP), false positives (corrections made where there was no error, FP), and false negatives (errors that were not corrected, FN). Racon and NextPolish do not report their corrections, so we omitted them from this comparison. (Note

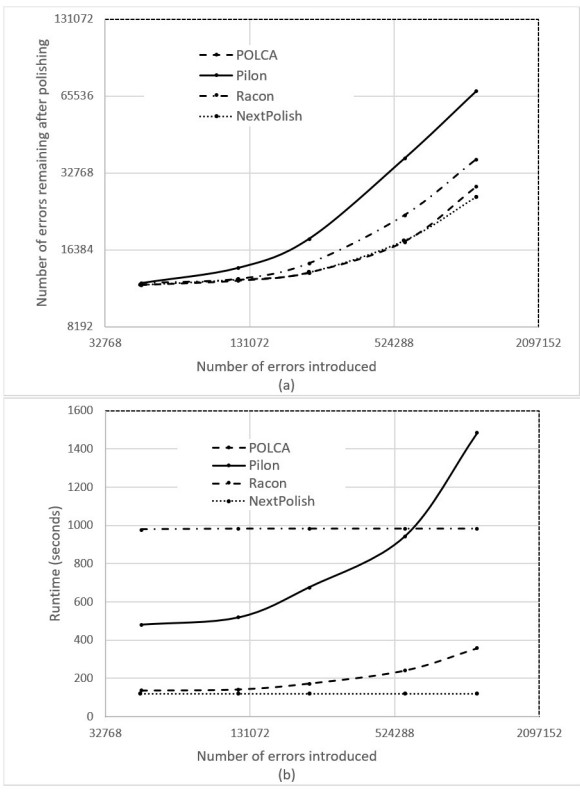

**Fig 1.** Upper panel (a) shows the results for POLCA, Pilon, Racon, and NextPolish in correcting simulated errors for five different experiments with different numbers of errors introduced into an assembly of the *Arabidopsis thaliana* genome. Lower panel (b) shows the running times (wall clock time) of each program, measured on a 16-core AMD Opteron system with 128Gb of RAM, running with 16 threads. The run times do not include the time spent on mapping the reads, which was the same for all programs.

that ntEdit [6] performed much worse than any of the other methods, so we did not include it in any of the details results shown here.) Table 1 shows that polishing with POLCA leaves a smaller number of total errors than Pilon across all three error rates. Pilon fixed more substitution errors than POLCA in all three experiments, and in one it fixed more insertion/deletion errors, but in both categories it also introduced many new errors, which resulted in an overall lower error rate for POLCA.

Racon and NextPolish do not provide base-by-base output, making it more challenging to compare corrections at the level of granularity shown in Table 1. Therefore, to evaluate Racon, Pilon, NextPolish and POLCA together, we used the Nucmer program from the MUMmer package [11] to align the polished sequence to the clean genome, and computed the alignment identity rate using the dnadiff software, also from the MUMmer package. We then estimated the number of bases in errors by multiplying this implied error rate by the clean genome size, 119,146,348 bp.

Fig 1A compares all four programs over the full range of simulated error rates. POLCA and NextPolish outperformed Racon and Pilon over the entire range of the error rates. POLCA was slightly better or equal to NextPolish for all error rates except for the highest, where NextPolish had a slight edge. POLCA and NextPolish were significantly faster than Pilon and Racon, as shown in Fig 1B. Note that here we only measured the time required for polishing, starting from the sorted, aligned reads, which were input to all three programs.

**Human data.**   We then evaluated the performance of the polishing techniques on a real data set, using a previously published assembly of the NA12878 human genome, GenBank accession GCA_001013985.1. That assembly was produced from PacBio SMRT data [10], and as such it was likely to contain more consensus-level sequence errors than an assembly based on Illumina data. Alignment of this assembly to the GRCh38.p12 human reference genome with nucmer, followed by dnadiff to compute differences, yields an average alignment identity rate of 99.66%.

For polishing this assembly, we used Illumina data for the same subject, NA12878, from the Genome In A Bottle project [12], dataset 140115_D00360_0009_AH8962ADXX, which contains 553,657,530 149-bp reads.

Because the "true" sequence of the NA12878 genome is not known, we evaluated, for each of the three polishing programs, whether the polished genome yielded a better alignment to the GRCh38.p12 sequence. The NA12878 assembly polished with POLCA had the closest alignment by a small margin, with 99.752% identity to GRCh38, while the assemblies polished with NextPolish, Pilon and Racon had 99.750%, 99.746% and 99.749% identity respectively. Thus all four polishing programs gave very similar results in terms of accuracy, however, POLCA and NextPolish ran considerably faster, completing the task in 4 hours and less than 1 hour respectively, while Racon took 15h 39m and Pilon took far longer, 150h 16m.

We note that Pilon is designed to do more than correct single base substitutions and short indel errors, which explains its longer run times. It attempts to identify and correct mis-assembled or collapsed repeats as well, a much more computationally demanding problem.

## Bacterial data

We tested the polishing approaches on four *Klebsiella pneumoniae* assemblies [1], for which all data as well as assemblies were made available at https://github.com/rrwick/Bacterial-genome-assemblies-with-multiplex-MinION-sequencing. We used Canu v1.5 [13] assemblies polished with Nanopolish [14] for isolates 1, 3, 4 and 5 as input to the polishing algorithms. The Canu assemblies were produced by the original authors [1] from Nanopore data alone and are available from the GitHub site. We used Illumina data from the corresponding isolates for

**Table 2. Polishing results for four *Klebsiella pneumoniae* isolates.** The columns list average identity rates for 1-to-1 best alignments of the polished assemblies to the finished sequences of the isolates. In bold we highlight the best result and any result within 0.01% of the best.

| Isolate barcode | Illumina coverage depth | Canu+Nanopolish Initial (%) | POLCA (%) | Pilon (%) | Racon (%) | NextPolish (%) |
|---|---|---|---|---|---|---|
| 01 | 60x | 99.62 | **99.96** | **99.96** | 99.76 | 99.94 |
| 03 | 38x | 99.01 | **99.89** | 99.86 | 99.88 | **99.90** |
| 04 | 44x | 99.79 | **99.93** | 99.89 | 99.88 | **99.94** |
| 05 | 68x | 99.35 | **99.98** | 99.97 | 99.68 | **99.98** |

polishing. The Illumina coverage depth for each isolate is shown in Table 2. We evaluated the polished assemblies by aligning them to the final, published sequences. The authors estimated that the error rates for those published sequences are below 0.00009%, i.e., the sequences are nearly perfect. We aligned the original Nanopore-only assemblies and the polished assemblies to the final sequences using MUMmer and then evaluated the average identity rate as described above for the Arabidopsis genomes. As shown in Table 2, POLCA performs as well as NextPolish and better than Pilon and Racon on these bacterial assemblies. All four programs improved the original (nanopore-only) assemblies substantially.

## Combining polishing tools

Because the programs use different algorithms for error correction, we ran an additional experiment to determine if users might benefit from running combinations of the programs on the same genome. Using simulated Arabidopsis data with a consensus error rate of 0.18%, we ran all combinations of two programs, in both orders, to polish the sequence. Table 3 compares the performance of the various pairs of polishing programs in this experiment. The fewest total errors were achieved by running POLCA followed by NextPolish. POLCA alone produced the fewest errors of any single program, however results were further improved by adding NextPolish to the protocol.

## Availability

POLCA is distributed freely under the GPLv3 license as part of the MaSuRCA genome assembly toolkit at https://github.com/alekseyzimin/masurca.

## Conclusion and future directions

POLCA provides an effective way to correct single-base substitution and short insertion/deletions errors in draft genome assemblies. On simulated data, it proved to be more accurate than Pilon and Racon and equivalent to the newer NextPolish method. POLCA was faster than Racon and Pilon, but slower than NextPolish. On simulated data, the most accurate polishing was achieved by using a combination of both POLCA and NextPolish. On real human and

**Table 3. Total number of erroneous bases (lower is better) remaining in the *Arabidopsis thaliana* genome with 231,929 introduced errors after polishing by two methods run consecutively.** The program shown in each row was run first, followed by the program shown in each column. The "Single run" column shows the number of errors remaining after a single run of each program. Note that in some cases the total number of errors increases after running two programs consecutively, such as after using Pilon or Racon on assemblies polished with NextPolish or POLCA.

| | Single run | POLCA | Pilon | Racon | NextPolish |
|---|---|---|---|---|---|
| **POLCA** | 13365 | 13363 | 13482 | 13961 | **13250** |
| **Pilon** | 18974 | 13360 | 16103 | 14217 | 13489 |
| **Racon** | 14646 | 14193 | 14686 | 14317 | 14194 |
| **NextPolish** | 13372 | 13372 | 13615 | 13721 | 13372 |

bacterial genome data, POLCA and NextPolish performed similarly, and better than Pilon and Racon, although POLCA appeared to be marginally better for human genome polishing. Our future plans for POLCA include continued maintenance to ensure the best performance with the latest sequencing data and speed improvements to stay competitive with the best available alternative software.

## Author Contributions

**Conceptualization:** Aleksey V. Zimin.

**Data curation:** Aleksey V. Zimin.

**Formal analysis:** Aleksey V. Zimin.

**Funding acquisition:** Aleksey V. Zimin, Steven L. Salzberg.

**Investigation:** Aleksey V. Zimin, Steven L. Salzberg.

**Methodology:** Aleksey V. Zimin.

**Software:** Aleksey V. Zimin, Steven L. Salzberg.

**Validation:** Aleksey V. Zimin.

**Visualization:** Aleksey V. Zimin.

**Writing – original draft:** Aleksey V. Zimin.

**Writing – review & editing:** Aleksey V. Zimin, Steven L. Salzberg.

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
