## [Decision Letter · Decision Letter 0]

15 Apr 2020

Dear Dr. Zimin,

Thank you very much for submitting your manuscript "The genome polishing tool POLCA makes fast and accurate corrections in genome assemblies" for consideration at PLOS Computational Biology. As with all papers reviewed by the journal, your manuscript was reviewed by members of the editorial board and by several independent reviewers. The reviewers appreciated the attention to an important topic. Based on the reviews, we are likely to accept this manuscript for publication, providing that you modify the manuscript according to the review recommendations.

Sincerely,

Christos A. Ouzounis

Associate Editor

PLOS Computational Biology

William Noble

Deputy Editor

PLOS Computational Biology

[LINK]

Reviewer's Responses to Questions

**Comments to the Authors:**

Reviewer #1: This manuscript describes a new pipeline for correcting long-read genome assemblies using short-read sequences. POLCA was compared against Pilon and Racon and demonstrated advantages over both. Comments below are provided to improve the utility of the manuscript.

I attempted to install MaSuRCA-v3.3.5 on a local workstation running openSUSE 15.0 but received a fatal error regarding the lack of the file 'xlocale.h'. Apparently this file was removed since glibc 2.26 so I had to form a symbolic link to '/etc/local.h'. The authors should fix this in the next distribution, or at least provide this information in the Github site. It seems that someone else already reported installation errors (issues #148 and #151 in Github). Lack of clean installs will limit the use of this tool.

The use of Freebayes for correction, and then batching for parallel correction, is very much like the pipeline described in the Vertebrate Genome Project (https://github.com/VGP/vgp-assembly/tree/master/pipeline/freebayes-polish). Are the authors affiliated with VGP? If so, then please credit this github repository. If not, please explain how POLCA is different from the VGP pipeline.

In the first paragraph of Introduction, please provide a reference to support the overall error rate of <1/100000 bases in hybrid assemblies.

On page 6 in the first paragraph of Simulated data experiments, the authors describe how the errors were introduced in the simulated Illumina reads (wgsim) - please describe how random errors were introduced in the clean genome so that one may reproduce the experiment.

When BWA maps an Illumina read from a repetitive region, doesn't it map the read to one of the repeats that is chosen at random, therefore isn't it possible for this read to provide incorrect correction at repetitive regions? This is an important drawback of Illumina correction of large genome assemblies so it would be useful to know how POLCA take this into account.

On page 10, Table 3: To help the table 'stand alone', it would be useful to know the original number of errors in the genome (perhaps in the Table header) and a column at the beginning to show the number of errors after a single run of each algorithm. The reader could more easily compare the utility and advantage of a second round of polishing.

Minor corrections

On page 6 in the first paragraph, the authors probably want to change '00.1%' to '0.1%'.

In the third line on page 7, 'smaller' instead of 'small'.

Reviewer #2: The authors present POLCA, a new genome polishing tool that used Illumina data to improve genome assemblies performed using long read technologies such as Nanopore and PacBio. They state that their newly developed tool performs faster than RACON and Pilon, the two most widely used tools, and with a comparable or higher accuracy. The authors mention three main advantages of POLCA: its speed, its low memory usage and its accuracy. To demonstrate this, they use three different examples: simulated data, human data and bacterial data. Finally, they assess the effect of using a combination of strategies.

The authors performed several tests to demonstrate their claims and thus the usefulness of their tool. The manuscript is well written and easy to follow. Also, their tool is easy to install and use. While POLCA is distributed as part of the MaSuRCA package it can also be used independently, thus it is a flexible tool. Additionally, the fact that it is distributed along with a widely used genome assembler will allow it reaching a large number of potential users.

Major Points:

While the authors have compared their tool with the most widely used tools, there are some more recent tools such as NextPolish or ntEdit which also claim to be faster than RACON and Pilon and similar accuracy levels. I consider that the manuscript would benefit from comparing the performance of some of these new tools with POLCA.

Minor Points:

The authors mention that “Whole-genome assemblies assembled using a hybrid sequence strategy can thereby obtain an overall error rate of less than 1 error per 100 thousand bases.” How have they calculated this number, or which is the reference used for this calculation?

The authors state that they empirically calculated the allele frequency threshold used by POLCA to be 2. I think that the manuscript would benefit from a more comprehensive description of the information and methodology used to determine the threshold.

**Have all data underlying the figures and results presented in the manuscript been provided?**

Reviewer #1: Yes

Reviewer #2: Yes

PLOS authors have the option to publish the peer review history of their article (what does this mean?). If published, this will include your full peer review and any attached files.

Reviewer #1: No

Reviewer #2: No
---

## [Decision Letter · Decision Letter 1]

25 May 2020

Dear Dr. Zimin,

We are pleased to inform you that your manuscript 'The genome polishing tool POLCA makes fast and accurate corrections in genome assemblies' has been provisionally accepted for publication in PLOS Computational Biology.

Best regards,

Christos A. Ouzounis

Associate Editor

PLOS Computational Biology

William Noble

Deputy Editor

PLOS Computational Biology

Reviewer's Responses to Questions

**Comments to the Authors:**

Reviewer #1: Revision is improved and should be published

Reviewer #2: The authors have addressed all of the comments and the updated results further highlight the usefulness of their tool. I would only suggest slightly modifying the abstract to reflect the extra work performed by the authors to compare POLCA with the newer polishers and not only with the two most popular ones.

**Have all data underlying the figures and results presented in the manuscript been provided?**

Reviewer #1: Yes

Reviewer #2: Yes

PLOS authors have the option to publish the peer review history of their article (what does this mean?). If published, this will include your full peer review and any attached files.

Reviewer #1: No

Reviewer #2: No

---

## [Editor Report · Acceptance letter]

12 Jun 2020

PCOMPBIOL-D-20-00106R1 

The genome polishing tool POLCA makes fast and accurate corrections in genome assemblies

Dear Dr Zimin,

I am pleased to inform you that your manuscript has been formally accepted for publication in PLOS Computational Biology. Your manuscript is now with our production department and you will be notified of the publication date in due course.

With kind regards,

Laura Mallard
